# Wanna Bet? Investigating the Factors Related to Adolescent and Young Adult Gambling

**Amelia Rizzo** [1,*], **Valentina Lucia La Rosa** [2], **Elena Commodari** [2], **Dario Alparone** [3], **Pietro Crescenzo** [4], **Murat Yıldırım** [5,6] **and Francesco Chirico** [7,8]

1    Department of Clinical and Experimental Medicine, University of Messina, 98122 Messina, Italy
2    Department of Educational Sciences, University of Catania, 95124 Catania, Italy; valarosa@unict.it (V.L.L.R.)
3    Department of Psychology, University of Rennes 2, 35700 Rennes, France
4    Department of Education Sciences, Psychology and Communication Sciences, University of Bari, 70121 Bari, Italy; pietro.crescenzo@uniba.it
5    Department of Psychology, Agri Ibrahim Cecen University, 04100 Agri, Turkey
6    Graduate Studies and Research, Lebanese American University, Beirut 1102-2801, Lebanon
7    Post-Graduate School of Occupational Health, Sacred Heart Catholic University, 75732 Rome, Italy
8    Health Service Department, Italian State Police, Ministry of the Interior, 20123 Milan, Italy
*    Correspondence: amrizzo@unime.it

**Abstract:** For many adolescents and young adults, gambling can represent an attractive and exciting form of entertainment, a way to take risks and explore new experiences. However, gambling also poses a significant risk for this age group. Research suggests that adolescents and young adults are particularly vulnerable to the negative consequences of gambling, including addiction, financial difficulties, and mental health issues. This paper aims to critically examine data on adolescent and young adult gambling, focusing on the risk factors associated with gambling behavior. A total of 326 subjects ($M_{age}$ = 19.01; $SD$ = 2.72), of whom 65.5% were female, completed a protocol containing a demographic sheet, the Lie/Bet Questionnaire, the Seven Domains Addiction Scale (7DAS), a subscale of the Addictive Behavior Questionnaire and the Coping Inventory for Stressful Situations (CISS). Specifically, we explored the impact of sociodemographic, environmental, psychological, and cognitive factors on adolescent and young adult gambling behavior. Clinical implications and future directions are discussed.

**Keywords:** adolescence; youth; gambling risk; coping; psychological experiences

## 1. Introduction

Adolescence and young adulthood are critical periods for developing several risk behaviors, including gambling [1–3]. Gambling is a popular activity worldwide, and the same can be said for adolescent gambling. The prevalence of adolescent gambling varies worldwide, with some studies indicating that up to 80% of adolescents have gambled at least once in their lifetime [4]. Furthermore, according to the European Network for Addictive Behavior study [5], the prevalence rate of adolescent gambling is alarmingly high, with up to 28.1% of adolescents participating in gambling activities regularly. In some countries, such as Spain and Italy, the prevalence of adolescent gambling is higher than in other countries, such as the United States and Canada [6].

Adolescents and young adults have been shown to be more vulnerable to the adverse effects of gambling compared to adults [7], as they are in a stage of brain development whereby decision-making and impulse control are still developing [8,9]. Thus, adolescents need to make informed decisions and clearly understand the potential risks involved with gambling [10,11]. The increased availability of online gambling has also increased the accessibility and prevalence of adolescent and young adult gambling and exposed them to more significant risks [12]. Indeed, online gambling has made it easier for adolescents to

access gambling activities without adult supervision or physical proximity to a gambling venue. Furthermore, online gambling has made it difficult for parents and educators to monitor and prevent gambling behaviors [13]. Online gambling provides easy access and anonymity and can be conducted from home, thus increasing the likelihood of adolescents and young adults engaging in gambling activities [14].

The consequences of excessive gambling can have severe and long-lasting effects on mental health and social functioning in this life stage [15]. Adolescents and young adults with gambling problems may experience increased negative mood states, leading to depression, anxiety, and suicidal ideation [16]. Furthermore, the financial burden of excessive gambling can lead to indebtedness and future social problems [17].

Studies have found a positive correlation between gambling and coping strategies in adolescence and young adulthood. For some individuals, gambling represents a coping mechanism to deal with stress and anxiety [18]. However, research has shown that this coping mechanism may exacerbate the negative consequences of stress and lead to problematic gambling behavior [19], especially in the case of previously experienced trauma, such as abuse, neglect, or mental health problems [20]. However, while gambling may temporarily distract or relieve stress, it can also lead to a cycle of compulsive behavior and further negative consequences [18].

Gambling in adolescence and young adulthood is a complex issue and various factors can influence it. One of the primary risk factors is a family history of gambling or addiction. In this regard, adolescents with parents or siblings with a history of gambling addiction are more likely to develop gambling problems themselves [21]. Another risk factor is peer pressure. In fact, youth with friends who gamble are more likely to gamble themselves, especially those already prone to risk-taking behaviors [22]. Furthermore, youth who have experienced trauma or stressful life events are also at an increased risk of developing a gambling addiction. Research suggests that adolescents who have experienced abuse or neglect, bullying, or family disruption are more likely to develop gambling problems [23].

Given the detrimental effects of gambling on adolescents' and young adults' mental and social health, the present study aimed to explore the relationships between psychopathological experiences, coping strategies, and problematic behaviors in a sample of adolescents and young adults. The study hypotheses can be formulated as follows:

**H1.** *Youth at risk of gambling report higher psychopathological experiences than healthy controls.*

**H2.** *Youth at risk of gambling employ more dysfunctional coping strategies than healthy controls.*

**H3.** *Youth at risk of gambling demonstrate more addictive and problematic behaviors than healthy controls.*

**H4.** *Psychopathological experiences, coping strategies, and problematic addictive behaviors predict youth gambling risk.*

## 2. Materials and Methods

### 2.1. Procedure

Data were collected through the Google Forms © platform from January 2023 to March 2023. The link was disseminated through mailing lists, social networks, and messaging apps to Italian-speaking high school and university students. Participation was voluntary, and anonymity was guaranteed, as requested by the ethical principles stated in the Declaration of Helsinki regarding subjects involved in research. Each participant, before completion, read and signed the online informed consent form, in which the purpose of the study was explained. In the case of underage subjects, online consent was provided by the parents. The present survey did not involve any manipulation, experimentation, or harmful effects on participants and followed the Guide to Internet Research Ethics Issued by the National Committee for Research Ethics in the Social Sciences and the Humanities (Published: 8/6/2019).

*2.2. Measures*

2.2.1. Lie-Bet Questionnaire

The Lie/Bet Questionnaire, adapted by Johnson et al. [24], is used to screen individuals for gambling disorders. It consists of two items: 1. "Have you ever felt the urge to play larger and larger amounts of money?"; 2. "Have you ever had to lie about how much you play to people close to you?". If the subjects answer no to both questions, they can still manage their gambling. If the subjects answer yes to at least one of these questions, they are in a dangerous situation concerning gambling (at risk), with a screening sensitivity of 100%. The Lie/Bet Questionnaire is a brief and easy-to-administer tool that can be used in various settings, such as primary care clinics, mental health clinics, and addiction treatment programs. It helps identify individuals at risk for gambling disorders and refer them to appropriate treatment services.

2.2.2. The Seven Domains Addiction Scale (7DAS)

The Seven Domains Addiction Scale (7DAS) is a subscale of the Addictive Behavior Questionnaire (ABQ) by Caretti et al. [25]. It explores seven different psychological domains identified in the literature as determinants for developing and maintaining pathological addiction. Each item was measured on a five-point Likert scale ranging from 0 (Never) to 4 (Always). The domains of 7DAS are as follows: Domain 1 (7 items): Separation anxiety, for the assessment of the subject's inability to adequately manage attachment relationships with significant figures (family members, friends, partners, etc.); e.g., "Do you ever feel like you are unable to establish satisfying intimate relationships?". Domain 2 (7 items): Affective dysregulation, to assess the subject's inability to identify, differentiate, and communicate emotions; e.g., "Do you ever find yourself feeling embarrassed or experiencing shame without understanding why?". Domain 3 (7 items): Somatoform and psychological dissociation, for the assessment of the presence of a sense of personal incoherence that leads the subject to manifest feelings of estrangement towards oneself, one's body, and others; e.g., "Do you ever feel your body, or a part of it, as if it were numb, paralyzed, or gone?". Domain 4 (7 items): Traumatic childhood experiences, for assessing the presence of traumatic memories related to experiences of neglect and emotional disavowal experienced at an early age; e.g., "As a child, did one or both of your parents make you feel unwanted?". Domain 5 (7 items): Impulse control, for assessing the inability to resist an impulse, compelling desire, or behavior, including dangerous or violent behavior; e.g., "Do you find yourself unable to wait your turn and experiencing distress during waits?". Domain 6 (7 items): Compulsive behavior and ritualization, for assessing the subject's tendency to enact repetitive, even dangerous, behavior compulsively; e.g., "Do you find yourself uncontrollably making purchases?". Domain 7 (7 items): Obsessive thoughts, for the assessment of excessive attention to the judgment of others on one's behavior; confusion and agitation when having to make a decision; recurrent and persistent thoughts that one cannot get rid of; e.g., "Do you find yourself constantly having thoughts that you just can't get rid of?". The 7DAS showed robust validity and reliability, with values of Cronbach's alpha ranging from 0.73 to 0.87 for each subscale [26]. In our sample, all subscales showed good internal consistency (Domain 1: $\alpha = 0.87$; Domain 2: $\alpha = 0.84$; Domain 3: $\alpha = 0.78$; Domain 4: $\alpha = 0.85$; Domain 5: $\alpha = 0.85$; Domain 6: $\alpha = 0.82$; Domain 7: $\alpha = 0.90$).

2.2.3. Coping Inventory for Stressful Situations (CISS)

The CISS is a self-report instrument for measuring coping, consisting of 48 items initially developed by Endler and Parker [27]. A version for adults and one for adolescents are available. The Italian version of the instrument, validated by Sirigatti and Stefanile [28], evaluates three coping strategies: task oriented (16 items), emotion oriented (16 items), and avoidance (16 items). The task-oriented strategy assesses the inclination to employ practical methods to solve problems by mentally reconfiguring the issue or trying to change circumstances and re-strategize one's actions. The emotion-oriented strategy includes emotional reactions, such as anger and blame, that are self-oriented and associated with

high levels of stress [27]. Avoidance describes activities and cognitive changes aimed at avoiding stressful situations and includes two subscales: distraction (avoiding stressful situations through distraction with other tasks), and social diversion (engaging in social activities to avoid stressful situations). Respondents are asked to rate each item on a five-point frequency scale ranging from 1 = not at all to 4 = very much; sample items of the scale are: "I focus on the problem and see how I can solve it." or "I feel anxious about my inability to handle the situation.". The questionnaire showed good psychometric characteristics of reliability and validity, with Cronbach's alpha value ranging from 0.71 to 0.86 for each subscale [28]. In this study, all subscales showed good internal consistency (Task-oriented: $\alpha = 0.90$; Emotion-oriented: $\alpha = 0.92$; Avoidance: $\alpha = 0.88$).

### 2.3. Statistical Analysis

Data were analyzed using SPSS 27.0 software (SPSS Inc., Chicago, IL, USA). Continuous variables were reported using the mean (M) and standard deviation (SD), whereas categorical variables were presented as frequencies and percentages. Correlations between continuous variables were evaluated using Pearson's correlation coefficient. Adolescents were divided into two groups according to the score on the Lie Bet Questionnaire (at risk vs. not at risk). For categorical variables, Pearson's Chi-square test was used to detect differences between groups. A one-way between-groups multivariate analysis of variance (MANOVA) was conducted with 7DAS and CISS scores as the dependent variables and risk of gambling as the between-subjects factor to investigate the differences in psychological domains involved in the development and maintenance of pathological addiction, as well as coping styles, between youth with and without gambling risk. Additionally, another one-way MANOVA was performed with risk behaviors scores as the dependent variables and risk of gambling as the between-subjects factor to explore the differences in adopting other risk behaviors between youth with and without gambling risk. Finally, a logistic regression model was developed to identify the primary risk factors for gambling behaviors in the young people in the sample. Gender, age, 7DAS psychological domains, coping styles, and risk behavior scores were included as independent variables, while the binary variable related to gambling risk was the dependent variable. Statistical significance was set at 0.05.

## 3. Results

### 3.1. Sample Characteristics

A total of 326 participants, ranging in age from 15 to 34 years (*M* = 19.01, *SD* = 2.72), of whom 65.5% were female, completed the survey. Subjects older than 25 (*N* = 12) were excluded from the inferential statistics, for a final sample of 314 valid cases. The sample size is appropriate. In more detail, according to the rule of thumb for logistic regression using six or more predictors, an absolute minimum of 10 participants per predictor variable is appropriate [29]. In this study, we have 14 predictor variables and consequently a required minimum sample size of 140. Sociodemographic variables were collected and analyzed separately by comparing the group of youth at risk of gambling behavior with the group without risk. Table 1 shows the sample characteristics stratified by groups. The Pearson's Chi-square test revealed no significant differences between groups, except for gender (*p* < 0.001).

**Table 1.** Sample characteristics by groups.

| | Gambling | | |
|---|---|---|---|
| | **At-Risk (*N* = 43)** | **Not at Risk (*N* = 283)** | ***p* Value** |
| **Gender** | | | |
| Male | 24 (55.8%) | 45 (15.9%) | <0.001 |
| Female | 19 (44.2%) | 238 (84.1%) | |
| **Age** | | | |
| Mean | 19.02 ± 2.41 | 19.01 ± 2.77 | |
| 15–19 years | 32 (74.4%) | 210 (74.2%) | |
| 20–25 years | 9 (20.9%) | 63 (22.3%) | 0.38 |
| 26–30 years | 2 (4.7%) | 4 (1.4%) | |
| 31–34 years | 0 (0.0%) | 6 (2.1%) | |
| **Physical pathology** | | | |
| Yes | 1 (2.3%) | 20 (7.1%) | 0.24 |
| No | 42 (97.7%) | 263 (92.9%) | |
| **Psychiatric diagnosis** | | | |
| Yes | 3 (7.0%) | 22 (7.8%) | 0.85 |
| No | 40 (93.0%) | 261 (92.2%) | |
| **Academic achievement** | | | |
| Under average | 0 (0.00%) | 1 (0.4%) | 0.77 |
| Around average | 11 (25.6%) | 60 (21.2%) | |
| Above average | 31 (72.1) | 212 (74.9%) | |
| **Massive Distance learning** | | | |
| Yes | 34 (79.1%) | 245 (86.6%) | 0.19 |
| No | 9 (20.9%) | 38 (13.4%) | |
| **Parents relationship** | | | |
| Separated | 8 (22.85%) | 44 (15.54%) | |
| Married | 35 (81.39%) | 223 (78.79%) | 0.44 |
| Cohabiting | 0 (0.00%) | 3 (01.06%) | |
| Single parent | 0 (0.00%) | 13 (04.59%) | |

Note. Results are presented as frequencies and percentages or mean ± standard deviation.

### 3.2. Correlations between Study Variables

A Pearson correlation analysis was conducted to examine the relationship between age, psychological domains of the 7DAS, and coping styles in youth with and without gambling risk (Table 2). In the group of adolescents with gambling risk, task-oriented coping was negatively correlated with impulse dyscontrol (r = −0.32, $p < 0.05$). Additionally, emotion-oriented coping was strongly positively correlated with all the 7DAS psychological domains ($p < 0.001$). Finally, avoidance-oriented coping was positively correlated with impulse dyscontrol (r = 0.362, $p < 0.05$). In the group of adolescents without gambling risk, emotion-oriented coping was strongly positively correlated with all the 7DAS psychological domains ($p < 0.001$). No significant correlations were found with the other coping styles.

**Table 2.** Correlations for study variables in youth with and without gambling risk.

| Variable | 1 | 2 | 3 | 4 | 5 | 6 | 7 | 8 | 9 | 10 | 11 |
|---|---|---|---|---|---|---|---|---|---|---|---|
| 1. Age | - | −0.001 | −0.007 | −0.023 | 0.124 * | 0.057 | 0.082 | 0.093 | 0.044 | −0.032 | −0.023 |
| 2. Separation anxiety | −0.007 | - | 0.736 *** | 0.541 *** | 0.362 *** | 0.658 *** | 0.601 *** | 0.691 *** | −0.063 | 0.546 *** | −0.034 |
| 3. Affect dysregulation | −0.132 | 0.740 *** | - | 0.624 *** | 0.305 *** | 0.752 *** | 0.652 *** | 0.762 *** | −0.070 | 0.581 *** | 0.015 |
| 4. Somatoform dissociation | −0.243 | 0.564 *** | 0.717 *** | - | 0.403 *** | 0.624 *** | 0.570 *** | 0.589 *** | 0.048 | 0.458 *** | 0.022 |
| 5. Childhood traumatic experiences | −0.174 | 0.335 * | 0.374 * | 0.422 ** | - | 0.445 *** | 0.510 *** | 0.353 *** | 0.024 | 0.300 *** | −0.085 |
| 6. Impulse dyscontrol | −0.184 | 0.741 *** | 0.750 *** | 0.706 *** | 0.370 * | - | 0.737 *** | 0.792 *** | −0.082 | 0.505 *** | 0.015 |
| 7. Compulsive behavior | −0.147 | 0.621 *** | 0.728 *** | 0.520 ** | 0.382 * | 0.725 *** | - | 0.723 *** | 0.068 | 0.451 *** | 0.095 |
| 8. Obsessive thoughts | −0.087 | 0.816 *** | 0.758 *** | 0.586 *** | 0.316 * | 0.802 *** | 0.770 *** | - | 0.040 | 0.556 *** | 0.037 |
| 9. Task-oriented coping | 0.181 | −0.123 | −0.067 | −0.096 | 0.241 | −0.318 * | −0.120 | −0.067 | - | 0.236 *** | 0.607 *** |
| 10. Emotion-oriented coping | −0.099 | 0.691 *** | 0.584 *** | 0.566 *** | 0.272 | 0.501 ** | 0.408 * | 0.643 *** | 0.312 * | - | 0.254 *** |
| 11. Avoidance-oriented coping | 0.172 | −0.104 | −0.043 | 0.036 | −0.169 | 0.362 * | −0.248 | −0.126 | −0.570 *** | 0.175 | - |

Note. Correlations for youth with gambling risk (n = 41) are reported below the diagonal. Correlations for youth without gambling risk (n = 273) are reported above the diagonal.
* $p < 0.05$. ** $p < 0.01$. *** $p < 0.001$.

### 3.3. Association between Psychological Domains, Coping Styles, and Gambling Risk

A MANOVA was conducted to analyze the differences in 7DAS psychological domains and coping styles between adolescents with and without gambling risk. Descriptive statistics were computed to summarize the means and standard deviations of each dependent variable for each group. Table 3 displays the results of the MANOVA analysis. The means and standard deviations of each dependent variable for both groups are reported. The F value, *p*-value, and partial eta-squared are also reported for each dependent variable. Results revealed a significant difference between the two groups on somatoform dissociation ($F$ = 6.71, $p$ = 0.010, partial eta-squared = 0.021) and impulse dyscontrol ($F$ = 4.41, $p$ = 0.036, partial eta-squared = 0.014). Therefore, adolescents with gambling risk would indeed be more likely to experience dissociative feelings and have lower impulse control. No significant differences were found in the coping styles adopted by the adolescents in the sample. Table 3 reports MANOVA results with descriptive statistics.

**Table 3.** MANOVA results with descriptive statistics.

| | Gambling | | | | |
| --- | --- | --- | --- | --- | --- |
| | At Risk (n = 41) | Not at Risk (n = 273) | *F* Value | *p* Value | Partial Eta-Squared |
| Separation Anxiety | 10.15 ± 7.03 | 10.07 ± 7.29 | 0.004 | 0.951 | 0.000 |
| Affect dysregulation | 11.64 ± 6.27 | 10.93 ± 6.17 | 0.387 | 0.534 | 0.001 |
| Somatoform dissociation | 6.54 ± 5.00 | 4.48 ± 4.25 | 6.708 | 0.010 | 0.021 |
| Childhood traumatic experiences | 2.54 ± 4.61 | 2.48 ± 3.91 | 0.005 | 0.946 | 0.000 |
| Impulse Dyscontrol | 11.39 ± 6.46 | 9.02 ± 6.09 | 4.412 | 0.036 | 0.014 |
| Compulsive behavior | 7.60 ± 5.51 | 7.29 ± 6.03 | 0.079 | 0.779 | 0.000 |
| Obsessive thoughts | 12.84 ± 7.42 | 12.67 ± 7.34 | 0.017 | 0.896 | 0.000 |
| Task-oriented coping | 38.60 ± 7.06 | 40.09 ± 8.40 | 0.949 | 0.331 | 0.003 |
| Emotion-oriented coping | 34.79 ± 9.59 | 36.08 ± 10.13 | 0.490 | 0.485 | 0.002 |
| Avoidance-oriented coping | 40.48 ± 7.45 | 39.81 ± 9.64 | 0.148 | 0.701 | 0.000 |

Note. Results are presented as mean ± standard deviation.

### 3.4. Association between Other Risk Behaviors and Gambling Risk

A MANOVA was conducted to analyze the differences in adopting a wide variety of risk behaviors between adolescents with and without gambling risk. Descriptive statistics were computed to summarize the means and standard deviations of each dependent variable for each group. As shown in Table 4, the comparison between the two groups showed a significant wide-ranging difference: adolescents at risk of gambling also reported a high number of risk behaviors in other areas such as alcohol use, heroin and opiates, cocaine, ecstasy, MDMA, cannabis, hashish or marijuana, poppers or inhalants. They also manifest an increased risk of substance-free addictive behaviors, such as spending a lot of time on pornographic sites, online role-playing games, video games, and gambling. Table 4. Comparison between adolescents with and without gambling on other risk behaviors.

### 3.5. Risk Factors for Gambling Behaviors in Youth

A logistic regression analysis was conducted to identify the main risk factors for gambling behaviors in the adolescents and young adults in the sample. The binary variable related to gambling risk was used as the dependent variable, while gender (0 = male; 1 = female), age, number of risk behaviors, and 7DAS psychological domain and coping style scores were used as independent variables. Table 5 reports the odds ratios of the logistic regression with the binary dependent variable gambling risk (0/1). The model was statistically significant ($\chi^2$ = 63.80, df = 14, $p$ < 0.001) and explained between 19.4% (Cox and Snell $R^2$) and 39.7% (Nagelkerke $R^2$) of the variance in gambling risk, and correctly classified 92.2% of cases. Gender ($p$ = 0.003), substance-free addictive behaviors ($p$ < 0.001), and childhood traumatic experiences ($p$ = 0.01) were significant predictors of gambling risk. Specifically, a high number of substance-free addictive behaviors (OR = 1.48, 95% CI [1.23, 1.77]) and the presence of childhood traumatic experiences (OR = 1.14, 95%

CI [0.99, 1.31]) increased the odds of gambling. Females had significantly lower odds of engaging in gambling behaviors than males (OR = 0.23, 95% CI [0.08, 0.60]).

**Table 4.** Comparison between adolescents with and without gambling on other risk behaviors.

|  | | Gambling | | | | |
|---|---|---|---|---|---|---|
|  | | At Risk (n = 41) | Not at Risk (n = 273) | *F* Value | *p*-Value | Partial Eta-Squared |
| **How often do you…?** | | | | | | |
| 1. | use alcohol? | 1.77 ± 1.04 | 1.29 ± 0.98 | 8.834 | **0.003** | 0.027 |
| 2. | use tobacco? | 1.51 ± 1.65 | 1.05 ± 1.48 | 3.471 | 0.063 | 0.011 |
| 3. | use coffee? | 2.23 ± 1.41 | 2.08 ± 1.51 | 0.397 | 0.529 | 0.001 |
| 4. | use heroin or other opiate derivatives? | 0.07 ± 0.26 | 0.01 ± 0.10 | 7.462 | **0.007** | 0.023 |
| 5. | use LSD or other hallucinogens? | 0.10 ± 0.29 | 0.01 ± 0.08 | 16.086 | **<0.001** | 0.047 |
| 6. | use ketamine or amphetamines? | 0.07 ± 0.26 | 0.01 ± 0.08 | 9.937 | **0.002** | 0.030 |
| 7. | use cocaine? | 0.09 ± 0.29 | 0.01 ± 0.08 | 15.869 | **<0.001** | 0.047 |
| 8. | use ecstasy or mdma? | 0.07 ± 0.25 | 0.01 ± 0.10 | 7.352 | **0.007** | 0.022 |
| 9. | use cannabis hashish or marijuana? | 0.69 ± 1.15 | 0.21 ± 0.59 | 17.014 | **<0.001** | 0.050 |
| 10. | use poppers or similar inhalants? | 0.07 ± 0.26 | 0.01 ± 0.08 | 10.072 | **0.002** | 0.030 |
| 11. | take psychotropic drugs or barbiturates? | 0.12 ± 0.32 | 0.10 ± 0.54 | 0.055 | 0.814 | 0.000 |
| 12. | spend a lot of time on the internet on social networks? | 2.74 ± 1.00 | 2.27 ± 1.07 | 7.525 | **0.006** | 0.023 |
| 13. | spend a lot of time on pornographic sites? | 0.86 ± 1.03 | 0.33 ± 0.68 | 19.095 | **<0.001** | 0.056 |
| 14. | spend a lot of time in online role-playing games? | 1.05 ±1.02 | 0.35 ±.814 | 25.391 | **<0.001** | 0.073 |
| 15. | spend a lot of time playing video games? | 1.09 ± 1.10 | 0.52 ± 0.92 | 13.362 | **<0.001** | 0.040 |
| 16. | gamble? | 1.51 ± 0.85 | 0.00 ± 0.00 | 898.895 | **<0.001** | 0.735 |

Note. Results are presented as mean ± standard deviation.

**Table 5.** Logistic regression analysis of gambling risk in the study sample.

|  | OR | 95% CI OR | | |
|---|---|---|---|---|
|  |  | Lower | Upper | *p* |
| Gender (Female) | 0.23 | 0.08 | 0.60 | **0.003** |
| Age | 0.98 | 0.81 | 1.18 | 0.84 |
| Number of substance-free addictive behaviors | 1.48 | 1.23 | 1.77 | **<0.001** |
| Number of substance addictive behaviors | 1.12 | 0.98 | 1.27 | 0.08 |
| Separation Anxiety | 1.03 | 0.93 | 1.14 | 0.54 |
| Affect dysregulation | 0.96 | 0.84 | 1.09 | 0.55 |
| Somatoform dissociation | 1.10 | 0.97 | 1.26 | 0.15 |
| Childhood traumatic experiences | 1.14 | 0.99 | 1.31 | **0.01** |
| Impulse Dyscontrol | 1.09 | 0.95 | 1.24 | 0.23 |
| Compulsive behavior | 0.99 | 0.87 | 1.12 | 0.86 |
| Obsessive thoughts | 1.00 | 0.88 | 1.13 | 0.94 |
| Task-oriented coping | 0.97 | 0.90 | 1.04 | 0.36 |
| Emotion-oriented coping | 0.94 | 0.89 | 1.00 | 0.07 |
| Avoidance-oriented coping | 1.04 | 0.98 | 1.11 | 0.19 |
| Constant | 0.13 | | | 0.41 |

Note. CI = Confidence Interval; OR = Odds Ratio.

## 4. Discussion

This study aimed to critically examine data on adolescent and young adult gambling, with a focus on the risk factors associated with gambling behavior. The comparative analysis between youth with and without gambling risk revealed the impact of sociodemographic, environmental, psychological, and cognitive factors on gambling behaviors in this developmental stage. Coping mechanisms, or the strategies that individuals use to manage stress and negative emotions, play a critical role in determining whether a youth is at risk for developing a gambling addiction.

Research on adolescent gambling suggests that coping strategies may be an important factor in understanding the relationship between gambling behavior and negative emo-

tional states [30]. In particular, individuals who experience high levels of stress, depression, or anxiety may be more likely to engage in gambling as a way to cope with negative emotions [31]. We found that, in the group of youth with gambling risk, task-oriented coping was negatively correlated with impulse dyscontrol. Additionally, emotion-oriented coping was strongly positively correlated with all the psychological domains. Finally, avoidance-oriented coping was positively correlated with impulse dyscontrol. Several studies have found a positive correlation between gambling and maladaptive coping strategies in adolescence. A study by Delfabbro and King [32] found that adolescents who engage in gambling as a form of coping have higher levels of problem gambling severity compared to those who use other coping strategies. It was also found that problem-gambling severity was positively associated with the use of avoidant coping strategies, such as denial, escape, and disengagement [33]. Another study by Blinn-Pike et al. [34] found that adolescents who engaged in gambling as a coping strategy reported higher levels of depression, anxiety, and stress compared to those who used other coping mechanisms. The study also found that adolescents who use gambling as a coping strategy showed a higher likelihood of developing gambling problems in the future. It has been demonstrated that adaptive coping strategies, such as problem-solving or seeking social support, may protect adolescents from developing gambling problems. In this regard, a study by Colder Carras and Kardefelt-Winther [35] found that adolescents who engaged in adaptive coping strategies were less likely to experience gambling problems, even when exposed to the same level of gambling opportunities and risks as those who engaged in maladaptive coping strategies.

We also found that youth with gambling risk are more likely to experience dissociative feelings and have lower impulse control. Dissociation refers to a mental process in which an individual disconnects from their surroundings, feelings, and even their own sense of identity [36]. Dissociation can be problematic for youth with gambling risk because it can negatively impact their decision-making abilities, which can lead to further risk-taking behavior [37,38]. In addition, lower impulse control can also contribute to increased gambling risk. Impulse control refers to an individual's ability to resist immediate gratification in order to achieve long-term goals [37]. Adolescents with lower impulse control may be more likely to engage in risky behaviors such as gambling, as they may have difficulty regulating their behavior and making thoughtful decisions about the potential risks and consequences of their actions [38].

Finally, findings showed that gambling risk is associated with more dependence both with and without substances. This result highlights the co-occurrence of addictive behavior among youth. This is a concerning finding, as it suggests that gambling risks could potentially be indicative of broader risk-taking behaviors and issues with impulse control [39]. The fact that adolescents at risk of gambling also exhibit an increased risk of substance-free addictive behaviors, such as spending a lot of time on pornographic sites, playing online role-playing games and video games, and gambling online, is particularly noteworthy. This finding highlights that problematic gambling behavior may be part of a broader pattern of addictive behaviors, which could negatively impact the individual's mental health and overall well-being [40,41].

Overall, our result suggests that individuals who have a high prevalence of addictive behaviors without substance use and have experienced childhood traumatic events are at a higher risk of developing gambling behaviors. Conversely, being female appears to be associated with a lower likelihood of engaging in gambling activities. These findings indicate that certain personal factors, such as gender and prior experiences, may play a role in the development of gambling behaviors. Research has shown that childhood adversity, such as physical, emotional, or sexual abuse, is significantly associated with adolescent gambling addiction [42]. Adolescents and young adults who experienced childhood adversity may turn to gambling as a way to cope with the negative effects of trauma. These experiences can lead to the development of a gambling addiction, which can exacerbate existing mental health issues, such as depression and anxiety. Understanding

the relationship between early life experiences and adolescent gambling addiction is crucial for early intervention and prevention.

*Limitations*

The present study has limitations that need to be acknowledged. First, the sample of subjects with gambling risk was limited, which may affect the generalizability of findings. Although the sample was recruited from various sources, it was a convenience sample and may not accurately represent the entire population of individuals at risk for problem gambling. Furthermore, the sample also had a higher proportion of females, which may limit the generalizability of findings to males. As there may be gender differences in the prevalence and expression of gambling problems, future studies should include an equivalent number of males and females to improve the representativeness of results. The subjects responded to an online survey. Hence, the lack of control by the experimenter could not limit the effects of errors due to possible confounding factors in the subject's environment. The effects of boredom, distraction, and social desirability were not estimated. Thus, caution must be taken when generalizing findings from this study to larger populations or individuals outside of the sample who have different characteristics.

## 5. Conclusions

Adolescent and young adult gambling addiction is a complex issue that can have significant negative consequences on mental health, social life, and academic performance. Youth who adopt maladaptive coping strategies may be more vulnerable to developing gambling problems, while those who use adaptive coping strategies may be more resilient to developing gambling problems.

The present study suggests that prevention and intervention strategies aimed at reducing adolescent gambling risk may benefit from focusing on developing effective coping skills that promote adaptive coping strategies. While specific psychological experiences may contribute to a youth's overall risk of developing a gambling addiction, the coping mechanisms that they use to manage those experiences lead to a greater risk. By promoting healthy coping mechanisms and addressing maladaptive coping strategies, we can help prevent the negative consequences of adolescent and young adult gambling and promote healthy emotional development. Additionally, parents and caregivers should be educated about gambling risks and encouraged to provide supportive environments [43] that promote healthy coping mechanisms. Overall, it is important to address the broader pattern of risk behavior exhibited by youth at risk of gambling, as this could be indicative of deeper underlying issues with impulse control and other mental health concerns [44–48]. Furthermore, early life experiences can have a significant impact on adolescent gambling addiction. Childhood trauma and exposure to gambling behaviors can lead to maladaptive coping mechanisms, including gambling, which can exacerbate existing mental health issues.

Targeted interventions could be critical in helping to mitigate the risk of negative outcomes associated with gambling and other addictive behaviors. Treating adolescent gambling addiction can be challenging, but psychodynamic, cognitive, and behavioral therapy models have been shown to be effective [49]. Prevention strategies, such as education and parental involvement, can also be effective in reducing the prevalence of adolescent gambling addiction. Future research should also evaluate the long-term effects of the recent COVID-19 pandemic on adolescents' and young adults' mental health [50,51] and on the increased risk of developing addictive behaviors such as gambling [52].

**Author Contributions:** Conceptualization, A.R. and V.L.L.R.; methodology, A.R. and V.L.L.R.; formal analysis, A.R., V.L.L.R. and E.C.; investigation, A.R., D.A. and P.C.; data curation, A.R. and V.L.L.R.; writing—original draft preparation, A.R. and V.L.L.R.; writing—review and editing, E.C., P.C., M.Y. and F.C.; visualization, A.R. and V.L.L.R.; supervision, E.C. and F.C. All authors have read and agreed to the published version of the manuscript.

**Funding:** This research received no external funding.

**Institutional Review Board Statement:** The present survey did not involve any manipulation, experimentation, or harmful effects on participants and followed the Guide to Internet Research Ethics Issued by the National Committee for Research Ethics in the Social Sciences and the Humanities (NESH) Published: 8/6/2019.

**Informed Consent Statement:** Informed consent was obtained from all subjects involved in the study.

**Data Availability Statement:** The datasets used and/or analyzed during the current study are available from the corresponding author upon reasonable request.

**Conflicts of Interest:** The authors declare no conflict of interest.

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
