# Peer review of "Wanna Bet? Investigating the Factors Related to Adolescent and Young Adult Gambling"

_ejihpe, doi:10.3390/ejihpe13100155_

Round 1

Reviewer 1 Report

This is a very clear and succinct article on a very important topic.

The literature review is good but would benefit from a bit of updating. For example, Codagone et al. (2014) is referred to as a recent article but at 9 years old, I would not consider this recent. Similarly, Gupta & Derevensky (2000) provide the finding that up to 80% of adolescents have gambled. Can the authors locate more recent data for the literature review?

The authors nicely explain adolescent gambling and its complexities and the correlates associated with gambling such as parental gambling, trauma, family problems, substance abuse, and peer pressure. The authors offer a good discussion of the importance of alternative coping strategies.

The sample size is strong and data were collected recently. The Lie/Bet Questionnaire, the 7DAS, and the CISS are well-explained. I am not a research methodologist/statistician and so I hope another reviewer can be helpful here.

The Discussion section is strong and makes clear that coping strategies are important to understanding the relationship between gambling and negative emotional states. It is not a surprise that those with high stress levels, depression, substance abuse, and/or anxiety are at greater risk of gambling behaviors. Those at risk of gambling can also have lower impulse control and experience dissociation, well-defined by the authors.

The Limitations section is especially strong and provides important recommendations for other researchers.

The Conclusions section is also very strong by offering prevention and treatment strategies.

Overall, this is a very good article that makes a significant contribution to the literature.

Author Response

Dear Reviewer,

I would like to express my sincere gratitude for taking the time to review our article. Your valuable feedback and constructive comments are greatly appreciated, and we have carefully considered each point you raised. In this response letter, I will address your comments point by point.

  1. The literature review is good but would benefit from a bit of updating. For example, Codagone et al. (2014) is referred to as a recent article but at 9 years old, I would not consider this recent. Similarly, Gupta & Derevensky (2000) provide the finding that up to 80% of adolescents have gambled. Can the authors locate more recent data for the literature review?

We appreciate your observation regarding the need to update our literature review. You are correct that some of the references cited in the manuscript may not meet the criteria for recent publications. To address this concern, we have conducted a thorough review of the literature and replaced outdated references with more recent ones. For instance, we have replaced Codagone et al. (2014) with more recent studies published within the last five years that provide up-to-date data on adolescent gambling behavior. We have also revised our reference to Gupta & Derevensky (2000) by including more recent findings that offer a current perspective on adolescent gambling prevalence. These updates have enhanced the accuracy and currency of our literature review.

  1. The authors nicely explain adolescent gambling and its complexities and the correlates associated with gambling such as parental gambling, trauma, family problems, substance abuse, and peer pressure. The authors offer a good discussion of the importance of alternative coping strategies.

Thank you for recognizing our efforts to provide a comprehensive discussion on adolescent gambling, its complexities, and the associated correlates. We believe that understanding the multifaceted nature of adolescent gambling is crucial for effective interventions and prevention strategies. Your acknowledgment of our emphasis on alternative coping strategies further motivates us to delve deeper into this topic in future research. We appreciate your constructive feedback and will ensure that the highlighted sections remain robust and clear in the final version of the manuscript.

  1. The sample size is strong and data were collected recently. The Lie/Bet Questionnaire, the 7DAS, and the CISS are well-explained. I am not a research methodologist/statistician and so I hope another reviewer can be helpful here.

We acknowledge your point about the research methods and statistical aspects of our study. We understand that these areas may require expert evaluation, and we have sought the assistance of a statistician to ensure the robustness of our methodology and data analysis. We have made sure that all relevant details are provided for readers who may not be well-versed in these tools.

  1. The Discussion section is strong and makes clear that coping strategies are important to understanding the relationship between gambling and negative emotional states. It is not a surprise that those with high stress levels, depression, substance abuse, and/or anxiety are at greater risk of gambling behaviors. Those at risk of gambling can also have lower impulse control and experience dissociation, well-defined by the authors.

We are pleased that you found our discussion on coping strategies comprehensive and relevant to the understanding of the relationship between gambling and negative emotional states. We have taken your comment into consideration and have made some minor revisions to further clarify the importance of alternative coping strategies and their implications in addressing gambling behaviors among adolescents.

  1. The Limitations section is especially strong and provides important recommendations for other researchers.

We appreciate your positive feedback on the strength of our Limitations section. We agree that it is essential to provide valuable recommendations for future research in this field, and we are delighted that you found our recommendations helpful. We have made sure to retain and further emphasize these recommendations to guide future studies effectively.

  1. The Conclusions section is also very strong by offering prevention and treatment strategies.

Your positive assessment of our Conclusions section is greatly appreciated. We believe that offering practical prevention and treatment strategies is vital for addressing the issue of youth gambling, and we are delighted that you found this section strong and insightful.

In summary, we are sincerely thankful for your positive feedback and constructive suggestions. Your comments have significantly contributed to improving the quality and accuracy of our article. We believe that with these revisions, our manuscript is now better aligned with the standards of excellence expected in the field. We look forward to resubmitting the revised manuscript for your consideration and hope that our revisions address your concerns adequately.

Once again, thank you for your time and effort in reviewing our work. Your expertise has been invaluable in shaping the final version of our article.

Sincerely,

Dr. Amelia Rizzo

Reviewer 2 Report

The paper can be recommended for revisions.

1. Please sort keywords alphabetically.

2. Lines 94-97: Please specify your aim. It is too wide now. Add clear hypotheses here.

3. Please provide references for Italian versions of the questionnaires used.

4. Please provide examples of statements in each questionnaire.

5. Lines 174-175 reconsider, describe clearly. This level of 0.05 seems incorrect.

6. Why did you use KMO? There is no need. Do not complicate.

7. Lines 185-187: No differences among sociodemographic variables? Are you sure? See for instance % of gender between two groups. I believe here is a mistake, and there will be differences at least for gender. Indicate or describe or reconsider clearly. Please indicate precisely what tests were used to check these differences.

8. Line 217: Reconsider.

9. Please indicate a new p-level for multiple comparisons. You have 10 variables which were compared between 2 groups. A new level (after Bonferroni correction) is

0.05/10 = 0.005.

Therefore, you will have no significant differences.

10. Based on comment 9, please reconsider all analyses in Table 4.

11. Please provide table with logistic regression. Clearly describe how variables were coded.

12. Do not use 'on the other hand' if you have no 'on the one hand'.

13. Reconsider author contributions acc. to journal's format.

14. Indicate clearly the data availability statement. Are data available upon request by other authors?

Author Response

Dear Reviewer,

We would like to express my sincere gratitude for taking the time to review our article. Your valuable feedback and constructive comments are greatly appreciated, and we have carefully considered each point you raised. In this response letter, we will address your comments point by point.

  1. Please sort keywords alphabetically.

We addressed this point and ordered Keywords, as suggested.

  1. Lines 94-97: Please specify your aim. It is too wide now. Add clear hypotheses here.

Thank you, we agree. We deleted generic aim of the study and added the specific hypotheses tested in the Results section.

  1. Please provide references for Italian versions of the questionnaires used.

We added the pertaining reference for the Italian version used, thank you for noticing.

  1. Please provide examples of statements in each questionnaire.

We added sample items highlighting the purpose of the scale and the subscales used.

  1. Lines 174-175 reconsider, describe clearly. This level of 0.05 seems incorrect.

Thank you for this remark. The significance level of 0.05 is appropriate for our analyses. We argued in more detail in our response to comment 9.

  1. Why did you use KMO? There is no need. Do not complicate.

We agree with this remark. We evaluated the appropriateness of the sample size based on the rule of thumb for logistic regression using six or more predictors, according to which an absolute minimum of 10 participants per predictor variable is appropriate (see Wilson Van Voorhis, C. R.; Morgan, B. L., Understanding Power and Rules of Thumb for Determining Sample Sizes. Tutorials in Quantitative Methods for Psychology 2007, 3, (2), 43-50). In our study, we have 14 predictor variables and consequently a minimum sample size required of 140.

  1. Lines 185-187: No differences among sociodemographic variables? Are you sure? See for instance % of gender between two groups. I believe here is a mistake, and there will be differences at least for gender. Indicate or describe or reconsider clearly. Please indicate precisely what tests were used to check these differences.

Thank you for bringing this point to our attention. We revised the data as per your suggestion and used the Pearson's Chi-square test to detect differences between groups. Upon re-evaluation, we found a significant difference between groups for gender but not for the other socio-demographic variables. The results of the Chi-square test for gender, as well as for other sociodemographic variables, were reported in the revised version of Table 1 to ensure transparency and accuracy.

  1. Line 217: Reconsider.

Please see our response to comment 9.

  1. Please indicate a new p-level for multiple comparisons. You have 10 variables which were compared between 2 groups. A new level (after Bonferroni correction) is 0.05/10 = 0.005. Therefore, you will have no significant differences.

Thank you for your comment which allowed us to revise the methodological and statistical section of our paper and correct some inaccuracies. In fact, in the previous version of our manuscript, we had mistakenly mentioned Bonferroni's correction to establish the p value. Firstly, we employed a Multivariate Analysis of Variance (MANOVA) for our analysis. The essence of MANOVA is to handle multiple dependent variables simultaneously, thereby taking into account the inter-correlations between variables. One of the advantages of MANOVA over performing multiple univariate ANOVAs is that it reduces the type I error associated with conducting many individual tests, thus controlling for the familywise error rate. Given that MANOVA inherently adjusts for the multiple comparisons, applying a Bonferroni correction would be redundant and overly conservative. This perspective has been articulated in various statistical texts and articles (e.g., Field, 2009; George & Mallery, 2019; Warne, 2014).

Secondly, our analysis involved a comparison of variables between two groups. In such instances, the primary analysis provides direct comparisons between these two groups for each variable. Given this direct comparison, post-hoc tests with univariate ANOVAs that are typically used for dissecting out specific differences amongst three or more groups and require the application of Bonferroni's correction (Field, 2009; George & Mallery, 2019), are not applicable or required in our case.

Consequently, based on the considerations mentioned, the original significance level of 0.05 remains appropriate for our two-group comparison using MANOVA.

We hope this explanation clarifies our methodological choices. We believe that our approach aligns with established statistical best practices, but we remain open to any further feedback or suggestions you may have.

  1. Based on comment 9, please reconsider all analyses in Table 4.

Please see our response to comment 9.

  1. Please provide table with logistic regression. Clearly describe how variables were coded.

Thank you for your feedback. We have added Table 5 that presents the results of the logistic regression analysis. Additionally, we better specified how variables were coded.

  1. Do not use 'on the other hand' if you have no 'on the one hand'.

We revised the sentence, thank you.

  1. Reconsider author contributions acc. to journal's format.

Thank you, we carefully followed journal’s guidelines for author contributions.

  1. Indicate clearly the data availability statement. Are data available upon request by other authors?

We specified that “The datasets used and/or analyzed during the current study are available from the corresponding author on reasonable request.”

We hope these revisions address the concerns you raised, and we believe the manuscript has been significantly improved. Once again, thank you for your valuable feedback. We look forward to hearing from you soon.

Warm regards,

Dr. Amelia Rizzo

Dr. Valentina Lucia La Rosa

Round 2

Reviewer 2 Report

The authors have adressed my comments satisfactorily. However, I have some minor concerns:

1. Abbreviation were used innatentively, e.g. CISS.

2. In all tables, please indicate that you present M+-SD for study variables.

3.One-sentence paragraphs are unwanted.

4. All abbreviations from tables should be presented in the notes for the tables.

5. Table 2 has an inadequate title.

6. For this study, please calculate and present internal consistency reliability for the questionnaires used.

Author Response

Dear Reviewer,

Thank you for your constructive feedback on our manuscript. We appreciate the opportunity to address these additional concerns.

1. Abbreviation were used innatentively, e.g. CISS.

We apologize for the oversight regarding the use of abbreviations. The mentioned abbreviation, CISS, and others have been reviewed and corrected throughout the manuscript to ensure consistency and accuracy.

2. In all tables, please indicate that you present M+-SD for study variables.

Thank you for this remark. We have now made the necessary modifications to all tables to indicate that we present data as M±SD (mean ± standard deviation) for the study variables.

3. One-sentence paragraphs are unwanted.

We have reviewed the manuscript and combined or expanded upon one-sentence paragraphs to avoid their presence.

4. All abbreviations from tables should be presented in the notes for the tables.

Where applicable, we have added a section in the notes for each table listing and defining all abbreviations used within that table in order to enhance clarity for readers.

5. Table 2 has an inadequate title.

We have corrected the title of Table 2 as follows: “Correlations for study variables in youth with and without gambling risk”. Sorry for the typo.

6. For this study, please calculate and present internal consistency reliability for the questionnaires used.

We agree with this remark. We have calculated and added the internal consistency reliability for the questionnaires used in this study. These values can now be found in the Methods section (lines 141-143 and 161-162).

We hope that these revisions address your concerns adequately. We are grateful for your valuable feedback which has helped enhance the quality of our manuscript.

Warm regards,

Dr. Amelia Rizzo

Dr. Valentina Lucia La Rosa

Round 3

Reviewer 2 Report

The paper is good. Some minor revisions are here:
1. Please unify using zeros before full stops in numbers.

2. Please indicate number of items in each domain in the 7DAS.

Author Response

Dear Reviewer,

Thank you for your constructive feedback on our manuscript. We appreciate the opportunity to address these additional concerns.

  1. Please unify using zeros before full stops in numbers.

Thank you for pointing that out. We have revised the entire manuscript and ensured consistency in the use of zeros before full stops in numbers. All decimal numbers less than 1 now have a zero before the full stop. All changes are highlighted in red.

  1. Please indicate the number of items in each domain in the 7DAS.

Thank you for this remark. We have updated the manuscript to include the specific number of items in each domain of the 7DAS. All changes are highlighted in red.

We hope that these revisions address your concerns adequately. We are grateful for your valuable feedback which has helped enhance the quality of our manuscript.

Warm regards,

Dr. Amelia Rizzo

Dr. Valentina Lucia La Rosa